# Recommending COVID-19 Vaccines to Patients: Practice and Concerns of Frontline Family Doctors

**DOI:** 10.3390/vaccines9111319

**Published:** 2021-11-13

**Authors:** Paul Kwok Ming Poon, Weiju Zhou, Dicken Cheong Chun Chan, Kin On Kwok, Samuel Yeung Shan Wong

**Affiliations:** 1JC School of Public Health and Primary Care, The Chinese University of Hong Kong, 2/F, Prince of Wales Hospital, Shatin, New Territories, Hong Kong, China; kwokmingpoon@cuhk.edu.hk (P.K.M.P.); joewjzhou@cuhk.edu.hk (W.Z.); dicken@cuhk.edu.hk (D.C.C.C.); kkokwok@cuhk.edu.hk (K.O.K.); 2Stanley Ho Centre for Emerging Infectious Diseases, The Chinese University of Hong Kong, Hong Kong, China; 3Hong Kong Institute of Asia-Pacific Studies, The Chinese University of Hong Kong, Hong Kong, China; 4Shenzhen Research Institute, The Chinese University of Hong Kong, Shenzhen 518057, China

**Keywords:** COVID-19, COVID-19 vaccine recommendations, family doctors, primary care

## Abstract

*Background*: Recommendation from doctors is a well-recognized motivator toward vaccine uptake. Family doctors are in the prime position to advise the public on COVID-19 vaccination. We studied the practice and concerns of frontline family doctors concerning COVID-19 vaccination recommendations to patients. *Methods*: We conducted a cross-sectional online survey of all family doctors in the Hong Kong College of Family Physicians between June and July 2021. Their practice of making COVID-19 recommendation to patients was assessed. Based on the Health Belief Model, factors associated with doctors’ recommendation practices were explored and examined. Multivariate logistic regression models were used to investigate the factors, including COVID-19 vaccine attributes, associated with doctors’ practices in making recommendations. Their own vaccination status and psychological antecedents to vaccine hesitancy were measured. *Results*: A total of 312 family doctors responded (a 17.6% response rate). The proportion of doctors who had received COVID-19 vaccines was 90.1%. The proportion of doctors who would recommend all patients without contraindications for the vaccination was 64.4%. The proportion of doctors who would proactively discuss COVID-19 vaccines with patients was 52.9%. Multivariate logistic regression analysis showed that doctors’ own COVID-19 vaccination status was the strongest predictor of family doctors making a recommendation to patients (aOR 12.23 95% CI 3.45–43.33). Longer duration of practice, willingness to initiate the relevant discussion with patients and less worry about vaccine side effects on chronic illness patients were the other factors associated with making a COVID-19 vaccination recommendation. *Conclusions*: Family doctors should be encouraged to get vaccinated themselves and initiate discussions with patients about COVID-19 vaccines. Vaccine safety data on patients with chronic illness, training and guidelines for junior doctors may facilitate the COVID-19 vaccination recommendation practices of family doctors.

## 1. Introduction

In the second year of the COVID-19 pandemic, countries are striving for a high vaccination coverage and herd immunity. However, since the rollout of vaccination programs worldwide, the overall coverage is still far from satisfactory, and there are marked inter-country and inter-regional disparities [1]. Equitable access to vaccines is a concern [2]. There are also many studies investigating vaccine hesitancy in different countries/regions [3]. Younger age, female sex, lower education level and income [4,5,6] are some common factors associated with lower vaccine uptake or acceptance. In addition, doctors’ recommendation is a consistent vaccine uptake facilitator found in studies [7] and across population subgroups including patients with chronic illness [8]. Indeed, a doctor’s recommendation is a well-recognized motivator toward the uptake of vaccines against other infectious diseases as well [9,10].

With the emergence of Variants of Concerns (VOC), especially the Delta variant [11,12], it is important to accelerate COVID-19 vaccination uptake worldwide. Even in countries that have attained relatively high coverages, the potential need for additional doses of new vaccines against VOC or further boosters would render their vaccination campaigns far from over. There is an urgent need for further research on facilitators of COVID-19 vaccines uptake. 

To date, COVID-19 vaccine studies involving doctors have mostly been on their own acceptance or hesitancy [13]. There is a paucity of research on factors associated with their practice or concerns about making COVID-19 vaccination recommendations to patients. Family doctors are at the forefront of community healthcare and in the prime position for providing advice to the public on COVID-19 vaccination. Therefore, we aimed to investigate factors associated with family doctors making COVID-19 vaccination recommendations to patients, and to identify targets for vaccine promotion interventions.

## 2. Methods

We conducted a cross-sectional online anonymous survey of family doctors in Hong Kong (HK). Participants provided informed consent via the survey platform. Participation was voluntary. We provided an incentive (HKD 50) to participants who had completed the survey. The survey was conducted between 17 June and 16 July 2021, around 4 months into the population-wide vaccination program launched by the HK government. 

### 2.1. Participants

Participants were members and fellows of the Hong Kong College of Family Physicians (HKCFP). The HKCFP is the sole governing body of the professional training of family medicine specialists in HK. The HKCFP’s fellows are trained doctors who obtained a fellowship in the HKCFP and/or a fellowship in the Royal Australian College of General Practitioners (FRACGP). The HKCFP’s members include trainees in family medicine and family doctors with or without postgraduate qualifications. In collaboration with the HKCFP, all its members and fellows (n = 1769) were invited to participate. 

### 2.2. Data Collection

#### 2.2.1. Primary Outcome

The primary outcome was respondents’ practice concerning recommending COVID-19 vaccines to patients, and was assessed by a question, “Will you recommend all your patients, who have no contraindications, for COVID-19 vaccination?”. The answer options included “yes”, “no” and “It’s hard to say, I will consider factors additional to contraindication”. A “yes” answer was considered a positive response. Other answers were considered a negative response.

#### 2.2.2. Considerations and Practice on Making COVID-19 Vaccination Recommendation

Respondents were asked if they would proactively discuss COVID-19 vaccination with patients. The attributes of vaccines that the respondents would consider when making a recommendation were also assessed. Respondents rated on a five-point Likert scale (1 = strongly disagree; 5 = strongly agree) their agreement on eight statements based on the Health Belief Model, including their perceived susceptibility and the severity of COVID-19 infections in their patients, the benefits (effectiveness), barriers (side effects), as well as cues to action (an additional laboratory test or clearer guidelines) to make a COVID-19 vaccination recommendation.

#### 2.2.3. General Vaccine Acceptance

Psychological antecedents of vaccine hesitancy were measured using a 13-item tool adapted from a “5C model” [14]. The five antecedents—including “confidence”, “complacency”, “constraints”, “calculation” and “collective responsibility”—were each assessed by two to three questions on a seven-point Likert scale (1 = strongly disagree; 7 = strongly agree). Mean scores under each antecedent were calculated. A higher score indicated stronger agreement on the antecedent. The 5C model has been used to study COVID-19 vaccine acceptance among healthcare workers and the general population in HK [15,16]. Views on the best timing for getting a COVID-19 vaccine were also explored. 

#### 2.2.4. Own COVID-19 Vaccination Status

Respondents’ own COVID-19 vaccine uptake was measured by their self-reported vaccination status. For those who had not gotten vaccinated, their intention to get vaccinated in the next 12 months was also solicited. Moreover, respondents rated the importance of their reasons on a four-point Likert scale (1 = not in my consideration at all; 4 = very important) regarding their decision to vaccinate (self-protection; protecting others; contributing to pandemic control; resumption of social life; acting as a role model for patients; encouraged in work environment; encouraged by the government) or not to vaccinate (doubtful safety; doubtful effectiveness; mild severity of the disease; own medical condition).

#### 2.2.5. Demographic and Practice-Related Variables

Demographic and practice-related variables—including gender, year of practice, type of practice, specialty, postgraduate qualifications, and provision of COVID-19 vaccination service in their clinic—were collected. 

### 2.3. Statistical Analysis

The primary outcome was analyzed as a binary variable. We described distributions of demographic and practice-related factors of respondents and examined their differences using a one-way ANOVA for continuous variables and a chi-squared test for categorical variables. Using a logistic regression model, we calculated crude odds ratios (ORs) and used 95% confidence intervals (CIs) to identify factors associated with making COVID-19 vaccination recommendations to patients. To minimize the possibility of overfitting in the multivariate logistic regression model, we adopted a forward stepwise method (probability for stepwise: entry *p* < 0.05, removal *p* > 0.1) to examine independent predictors for making vaccination recommendations. All statistical analyses were performed using the SPSS Statistical package (Windows version 26.0; SPSS Inc., Chicago, IL, USA).

## 3. Results

A total of 312 respondents completed the survey. The response rate was 17.6%, which is comparable to other online surveys conducted among doctors [17]. The proportion of 51.3% of respondents were male. A proportion of 47.4% had practiced for over 20 years, 34.3% had practiced for 10–20 years and 18.3% had practiced for fewer than 10 years. A proportion of 52.6% worked in the public healthcare sector, and 47.1% worked in the private sector. A proportion of 42.9% were family medicine specialists, and 82.1% had at least one postgraduate qualification. A proportion of 62.5% provided a COVID-19 vaccination service in their clinics. In a univariate analysis, longer years of practice (>20 years: OR 8.96, 95% CI 4.49–17.89), providing a COVID-19 vaccination service at their clinic (OR 2.34, 95% CI 1.45–3.77), having a postgraduate qualification (OR 2.08, 95% CI 1.16–3.74) and practicing in the private sector (OR 1.67, 95% CI 1.04–2.68) were associated with making recommendations (Table 1).

A proportion of 90.1% had received COVID-19 vaccines themselves, 64.4% would recommend all patients without contraindications for COVID-19 vaccination, and 52.9% would proactively discuss COVID-19 vaccines with patients. Among vaccinated doctors, 70.1% would recommend all patients without contraindication for vaccination and 55.5% would proactively discuss the issue with patients (Table 2). In addition to the side effects (92.0%) and efficacy (90.4%), the availability of Phase III clinical trial data was the vaccine attribute that most (75.3%) respondents would consider when making a recommendation. A proportion of 88.1% agreed that COVID-19 vaccines are effective in preventing the infection of patients, and 87.2% agreed that the vaccines are safe for the majority of their patients. However, up to 36.2% said that they were worried about serious side effects of the vaccines on patients with chronic illnesses (Table 3).

In multivariate analysis (Table 4), the strongest independent predictor for making recommendation was vaccination of the doctors themselves (aOR 12.23, 95% CI 3.45–43.33). Nevertheless, there were still around one-third of the vaccinated doctors who would not recommend all patients without contraindication for the vaccination. Doctors who would proactively discuss the issue with their patients were more likely to make recommendation (aOR 3.62, 95% CI 1.84–7.14). “Year of practice” was the only demographic or practice-related factor independently associated with making recommendation (>20 years: aOR 3.55, 95% CI 1.49–8.44) and providing COVID-19 vaccination at clinic was no longer a significant factor in multivariate analyses. Doctors who would consider availability of Phase III trial data (aOR 0.38, 95% CI 0.16–0.88) and vaccine type (aOR 0.44, 95% CI 0.22-0.85) were less likely to make recommendation. Doctors with less worry about serious vaccine side effects on patients with chronic illness were more likely to make recommendation (“Neutral” aOR 2.49, 95% CI 1.14–5.45; “Disagree” aOR 3.59, 95% CI 1.72–7.47). Regarding antecedents on vaccine hesitancy measured by the “5C model”, only “confidence” was associated with making recommendation (aOR 1.37, 95% CI 1.03–1.83).

## 4. Discussion

### 4.1. Summary

Our study showed that the high COVID-19 vaccination rate of family doctors did not lead to the same high proportion of family doctors making vaccination recommendations to patients. Nevertheless, own COVID-19 vaccination status was still the strongest predictor of family doctors making recommendations to patients. Longer practice, proactive discussion with patients about vaccines and less worry about vaccines’ side effects on chronically ill patients were the other factors associated with making recommendations. 

### 4.2. Strengths and Limitations

To the best of our knowledge, this is the first study to examine factors associated with family doctors making COVID-19 vaccination recommendations to their patients. Our sampling frame was comprehensive and covered the vast majority of family doctors in our locality. This study has limitations. Firstly, there could be a risk of a selection bias, with those being more vaccine-conscious more likely to provide a response. Moreover, the overall COVID-19 vaccination coverage in HK was nearly 50%, which was higher than that in most countries [1]. As such, there could be an overestimation of the vaccine coverage among the doctors and the proportion who would make recommendations to patients. Nevertheless, this would only mean that the actual situation in other regions could be worse and would call for more intense interventions. Second, there could be a risk of duplicated responses in an anonymous survey. We minimized that risk by crossmatching the email addresses of respondents for de-duplication, our reminder emails were only sent to those who had not responded, and recipients were also asked not to respond twice. Third, psychological antecedents of vaccine hesitancy were measured by a modified 5C model, with two items in the original model under the antecedents “confidence” and “constraint” removed based on the local context and the fact that our respondents were doctors themselves. Our results might only provide a clue about the relationship between doctors’ own vaccine hesitancy and their practice concerning vaccine recommendation to patients, and more research on the association is needed. Fourth, the response rate was not high. Multiple factors including the busy schedules of doctors and lower response rate in online surveys (as opposed to mailed survey) [18] may be the reasons. There could be differences in the factors associated with vaccination recommendation between respondents and non-respondents that might have introduced bias. Nevertheless, the response rate is comparable to, or slightly better than, other online surveys conducted overseas among doctors [17], or locally among doctors [19] or general public [20].

### 4.3. Comparing with Existing Literature

Our results showed that COVID-19 vaccination coverage among family doctors in HK was high with 9 out of 10 (90.1%) having already received the vaccines (there was a further 9% planning to receive the vaccine in the next 12 months). This is higher than the range of 27.7% to 78.1% COVID-19 vaccines acceptance among doctors and nurses reported in various countries [3] and a 75.1% coverage among physicians and other healthcare providers reported in a US study [21]. Regarding reasons of getting vaccinated themselves, “protecting family, colleagues or patients” and “self-protection” were considered very important by 79.4% and 73.4% respondents respectively. However, not all vaccinated family doctors would make COVID-19 vaccination recommendation to their patients. There were only 64.4% of respondents who would recommend all patients without contraindications for the vaccination, and only 52.9% would proactively discuss with patients on COVID-19 vaccines. Having said that, “having received COVID-19 vaccines themselves” was still the strongest independent predictor of making recommendation to their patients (aOR 12.23). Pauline Paterson et al. [10], in a systematic review, also found that healthcare providers were more likely to recommend vaccination if they were themselves vaccinated and the association was observed for different vaccines including human papilloma virus and influenza vaccines. Nevertheless, there are clearly other factors to tackle in order to further engage them in vaccine promotion to the public. As the first step, frontline doctors should be encouraged to initiate a discussion with their patients on COVID-19 vaccination, and we found that doctors willing to start the discussion were more likely to make the recommendation to their patients. 

While vaccine efficacy and safety are known to be important factors in the consideration of healthcare workers to get vaccinated themselves [22], our study showed that there are additional considerations when it came to making medical advice or professional recommendations to patients. Indeed, nearly 90% of our respondents agreed that, in general, COVID-19 vaccines are effective in preventing the infection and are safe for the majority of their patients. However, it turned out that worries of serious side effects on certain patient groups (i.e., those with chronic illness) determined their practice in making recommendation and we also observed an apparent dose-response relationship between degree of the worry and practice concerning recommendations (Table 4). Without long post-market safety surveillance, and with wide media coverage on serious cases of adverse events following immunization, there were worries among the local community and doctors regarding side effects of the new COVID-19 vaccines, particularly in patients with chronic illnesses. In response, the local government had also issued interim guidance notes for doctors to address the uncertainties [23]. As such, general efficacy and safety information on COVID-19 vaccines may not be sufficient to put doctors at ease about making recommendations, and more in-depth safety data on patients suffering with chronic illnesses are needed. This was also reflected in our results that the consideration of the availability of Phase III trial data on vaccines was an independent predictor on making recommendations. Furthermore, a previous study showed that data on post-licensure safety monitoring would also be influential on doctors’ confidence in new vaccines [24]. Furthermore, patients with chronic illnesses are a vulnerable group for serious complications of COVID-19 infections [25]. Apart from the provision of data to doctors, building confidence in COVID-19 vaccines among patients with chronic illnesses via education about vaccine safety would be equally important.

Our results also revealed that family doctors with longer years of practice were more likely to make COVID-19 vaccination recommendations than junior doctors. The experience of senior doctors could make them more confident in handling the controversies and uncertainties surrounding COVID-19 vaccination. This highlights the need for training, clinical aids or guidelines for clinical assessment of vaccine suitability, especially for the less experienced doctors. It is noteworthy that the need for clearer guidelines and additional laboratory tests were associated with making recommendations in our univariate analysis, although they were not statistically significant in multivariate analyses. In Hong Kong, the government has been constantly updating the clinical reference materials for local doctors on the efficacy and safety profiles of COVID-19 vaccines [26]. A previous study [27] revealed inter-country discrepancies in the proportion of people receiving a doctor’s recommendation for COVID-19 vaccination, with 31.9% and 50.2% reported by respondents in China and the US, respectively. We demonstrated that there could be multiple factors behind these discrepancies and further research on interventions to enhance the engagement of frontline doctors in COVID-19 vaccine promotion campaigns is needed.

### 4.4. Implications for Research and Practice

This survey revealed that family doctors vaccinated against COVID-19 would be more likely to make vaccination recommendations to their patients. This highlights, in addition to self-protection, another important aspect of COVID-19 vaccination among frontline family doctors. Public health authorities and governments should continue to encourage frontline family doctors—who are a high-risk group for serious COVID-19 infections [28]—and, at the same time, key players in public vaccine promotion, to get vaccinated themselves. To further increase the likelihood of them making vaccination recommendations, frontline doctors should be encouraged to initiate discussions with their patients about COVID-19 vaccination. Further studies may look into family doctors’ knowledge of, and abilities to discuss with patients, the new technologies involved in the production of various COVID-19 vaccines.

We also found that longer practice and less worry about vaccines’ side effects on chronically ill patients were associated with making recommendations to patients. This suggests that more vaccine safety data on patients with chronic illness, training and guidelines for the assessment of patients’ COVID-19 vaccine suitability are needed, especially for junior doctors.

## 5. Conclusions

Having received the vaccines themselves is the strongest predictor of frontline family doctors making COVID-19 vaccination recommendations to patients. Doctors should also be encouraged to initiate discussions with patients about the vaccines. More vaccine safety data on patients with chronic illness, training and guidelines for the assessment of patients’ COVID-19 vaccine suitability are needed, especially for junior doctors. 

## Figures and Tables

**Table 1 vaccines-09-01319-t001:** Descriptive statistics of demographic and practice-related characteristics of family doctors and their association with making COVID-19 vaccination recommendations (N = 312).

	All Respondents		Recommend Vaccination to All Patients without Contraindications							
			No *		Yes					
Variable	n = 312 (%)		n = 111 (%)		n = 201 (%)		*p* ^~^	Crude OR	95% CI	
**Gender**										
Male	160	51.3	50	45.0	110	54.7	0.101	Ref		
Female	152	48.7	61	55.0	91	45.3		0.68	0.43	1.08
**Postgraduate qualification** (any)										
No	56	17.9	28	25.2	28	13.9	0.013	Ref		
Yes	256	82.1	83	74.8	173	86.1		**2.08** ^#^	**1.16**	**3.74**
**Specialty**										
*Family medicine specialist ^@^*										
*No*	178	57.1	69	62.2	109	54.2	0.175	Ref		
*Yes*	134	42.9	42	37.8	92	45.8		1.39	0.86	2.23
*General practitioner (i.e., a non-specialist) with a postgraduate qualification in family medicine (e.g., diploma)*										
*No*	187	59.9	64	57.7	123	61.2	0.542	Ref		
*Yes*	125	40.1	47	42.3	78	38.8		0.86	0.54	1.38
**Type of practice**										
Public	164	52.6	67	60.4	97	48.3	0.033	Ref		
Private	147	47.1	43	38.7	104	51.7		**1.67** ^#^	**1.04**	**2.68**
Others	1	0.3	1	0.9	0	0.0		-	-	-
**Year of practice**										
<10 years	57	18.3	38	34.2	19	9.5	<0.001	Ref		
11–20 years	107	34.3	46	41.4	61	30.3		**2.65** ^^^	**1.36**	**5.19**
>20 years	148	47.4	27	24.3	121	60.2		**8.96** ^&^	**4.49**	**17.89**
**Offered a COVID-19 vaccination service in their clinic**										
No	117	37.5	56	50.5	61	30.3	<0.001	Ref		
Yes	195	62.5	55	49.5	140	69.7		**2.34** ^^^	**1.45**	**3.77**

* A total of 100 participants answered, “It’s hard to say, I will consider factors additional to contraindication”. ^~^ Chi-squared test *p*-values. ^@^ The doctors had undertaken structured postgraduate vocational training from the Hong Kong College of Family Physicians. The “Others” group was not included in calculating *p*-values due to its small number (n = 1; i.e., “unemployed”). ^#^
*p* < 0.05; ^ *p* < 0.01; ^&^
*p* < 0.001. Significant ORs (95% CI) are presented in bold.

**Table 2 vaccines-09-01319-t002:** Vaccination status of family doctors and their practice concerning recommendation and discussion with patients about COVID-19 vaccines (N = 312).

	All Participants		Have Vaccinated		Will Vaccinate in Next 12 Months		Not Vaccinate	
Variable	n = 312 (%)		n = 281 (%)		n = 28 (%)		n = 3 (%)	
**Recommend all patients without** **contraindications for vaccination**								
Yes	201	64.4	197	70.1	4	14.3	0	0.0
No	111	35.6	84	29.9	24	85.7	3	100.0
**Proactively discuss vaccination with patients**								
Yes	165	52.9	156	55.5	9	32.1	0	0.0
No	147	47.1	125	44.5	19	67.9	3	100.0

**Table 3 vaccines-09-01319-t003:** Descriptive statistics and univariate analyses of factors associated with making COVID-19 vaccination recommendations to patients among family doctors (N = 312).

	All Respondents		Recommend Vaccinations to All Patients without Contraindications						
			No *		Yes				
Variable	n = 312 (%)		n = 111 (%)		n = 201 (%)		Crude OR	95% CI	
**Have received COVID-19 vaccine**									
No	31	9.9	27	24.3	4	2.0	Ref		
Yes	281	90.1	84	75.7	197	98.0	**15.83** ^&^	**5.37**	**46.65**
**Proactively discuss with patients**									
No	147	47.1	80	72.1	67	33.3	Ref		
Yes	165	52.9	31	27.9	134	66.7	**5.16** ^&^	**3.11**	**8.58**
**Attributes of vaccine to consider when making recommend**									
*Availability of Phase III clinical trials data*									
*No*	77	24.7	14	12.6	63	31.3	Ref		
*Yes*	235	75.3	97	87.4	138	68.7	**0.32** ^&^	**0.17**	**0.60**
*Vaccine efficacy*									
*No*	30	9.6	7	6.3	23	11.4	Ref		
*Yes*	282	90.4	104	93.7	178	88.6	0.52	0.22	1.26
*Side effects*									
*No*	25	8.0	5	4.5	20	10.0	Ref		
*Yes*	287	92.0	106	95.5	181	90.0	0.43	0.16	1.17
*Approved by local government*									
*No*	141	45.2	66	59.5	75	37.3	Ref		
*Yes*	171	54.8	45	40.5	126	62.7	**2.46** ^&^	**1.53**	**3.96**
*Listed by WHO for emergency use*									
*No*	163	52.2	62	55.9	101	50.2	Ref		
*Yes*	149	47.8	49	44.1	100	49.8	1.25	0.79	2.00
*Vaccine type (e.g., inactivated vaccine, mRNA)*									
*No*	151	48.4	54	48.6	97	48.3	Ref		
*Yes*	161	51.6	57	51.4	104	51.7	1.02	0.64	1.62
*Manufacturer*									
*No*	202	64.7	68	61.3	134	66.7	Ref		
*Yes*	110	35.3	43	38.7	67	33.3	0.79	0.49	1.28
**Consider susceptibility of individual patient when making recommendation**									
Agree	192	61.5	75	67.6	117	58.2	Ref		
Neutral	63	20.2	27	24.3	36	17.9	0.86	0.48	1.52
Disagree	57	18.3	9	8.1	48	23.9	**3.42** ^^^	**1.59**	**7.37**
**Consider severity of COVID in individual patient when making recommendation**									
Agree	64	20.5	31	27.9	33	16.4	Ref		
Neutral	42	13.5	26	23.4	16	8.0	0.58	0.26	1.28
Disagree	206	66.0	54	48.6	152	75.6	**2.64** ^^^	**1.48**	**4.72**
**There are insufficient data to support recommendation**									
Agree	97	31.1	46	41.4	51	25.4	Ref		
Neutral	93	29.8	40	36.0	53	26.4	1.20	0.67	2.12
Disagree	122	39.1	25	22.5	97	48.3	**3.50** ^&^	**1.93**	**6.34**
**Worried of serious side effects in patients with chronic illness**									
Agree	113	36.2	65	58.6	48	23.9	Ref		
Neutral	67	21.5	25	22.5	42	20.9	**2.28** ^^^	**1.22**	**4.23**
Disagree	132	42.3	21	18.9	111	55.2	**7.16** ^&^	**3.94**	**13.01**
**Need additional laboratory test for making recommendation**									
Agree	149	47.8	64	57.7	85	42.3	Ref		
Neutral	79	25.3	30	27.0	49	24.4	1.23	0.70	2.15
Disagree	84	26.9	17	15.3	67	33.3	**2.97** ^^^	**1.59**	**5.54**
**Need clearer guidelines for making recommendation**									
Agree	128	41.0	66	59.5	62	30.8	Ref		
Neutral	78	25.0	23	20.7	55	27.4	**2.55** ^^^	**1.40**	**4.63**
Disagree	106	34.0	22	19.8	84	41.8	**4.07** ^&^	**2.27**	**7.29**
**5C model**									
Confidence	5.58	1.17	5.13	1.13	5.82	1.11	**1.69** ^&^	**1.37**	**2.08**
Complacency	1.97	0.93	2.26	1.00	1.82	0.85	**0.60** ^&^	**0.46**	**0.77**
Constraints	2.11	1.27	2.50	1.34	1.90	1.19	**0.69** ^&^	**0.57**	**0.83**
Calculation	6.37	0.82	6.40	0.63	6.35	0.91	0.93	0.69	1.24
Collective responsibility	4.87	0.77	4.89	0.80	4.86	0.75	0.96	0.70	1.30

* 100 participants answered “It’s hard to say, I will consider factors additional to contraindication”. ^ *p* < 0.01; ^&^
*p* < 0.001. Significant ORs (95% CI) are presented in the bold numbers.

**Table 4 vaccines-09-01319-t004:** Stepwise logistic regression * to identify independent predictors of making COVID-19 vaccination recommendations to patients by family doctors (N = 312).

Variable	aOR ^#^	95% CI		*p*
**Years of practice**				
<10 years	Ref			
11–20 years	1.20	0.53	2.71	0.663
>20 years	**3.55**	**1.49**	**8.44**	**0.004**
**Have received vaccine**				
No	Ref			
Yes	**12.23**	**3.45**	**43.33**	**<0.001**
**Proactively discuss with patients**				
No	Ref			
Yes	**3.62**	**1.84**	**7.14**	**<0.001**
**Attributes of COVID-19 vaccine to consider when making recommendation**				
*Availability of Phase III clinical trial data*	**0.38**	**0.16**	**0.88**	**0.024**
*Vaccine type (e.g., inactivated vaccine, mRNA)*	**0.44**	**0.22**	**0.85**	**0.015**
**Worry about the side effects on patients who have chronic illnesses**				
Agree	Ref			
Neutral	**2.49**	**1.14**	**5.45**	**0.023**
Disagree	**3.59**	**1.72**	**7.47**	**0.001**
**5C model**				
Confidence	**1.37**	**1.03**	**1.83**	**0.031**

* The model included all variables in Table 1 and Table 3. ^#^ Adjusted odds ratios. Significant aORs (95% CI) are presented in bold.

## Data Availability

All data generated or analyzed during this study are included in this published article.

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
