# Peer review of "Recommending COVID-19 Vaccines to Patients: Practice and Concerns of Frontline Family Doctors"

_vaccines, 2021, doi:10.3390/vaccines9111319_

Round 1

Reviewer 1 Report

The present study illustrates the factors that affect the family doctors attitude on recommending COVID-19 vaccination to their patients in Hong Kong. The study was well organised and the results are also well presented. The authors should revise the use of the English language in several parts of the manuscript (e.g. 187-188, 194-195, 221, 273.

Lines 54-55: "..the importance of accelerating COVID-19 vaccination uptake worldwide cannot be overemphasized?"

Lines 131 and elsewhere: Please, try to use spelled-out numbers at the beginning of a sentence in place of numerals. This distinction is not based on grammar, but rather the conventions of academic writing in English. You can also try to reword the sentence.

Line 131 and elsewhere: please avoid the use of the word "done"....you can use the word "conducted", "carried out"

It would be of great interest if the family doctors could  provide information whether they could discuss with their patients the new technology of the COVID19 vaccines (mRNA, cDNA -genetic engineering)

Reviewer 2 Report

The original "Recommending COVID-19 vaccines to patients: practice and concerns of frontline family doctors" is reviewed and its acceptance with minor revisions is recommended, based on the following sections, since it provides new information on some important aspects:
In line 108 and in line 213 it is indicated that those already vaccinated and those who intend to be vaccinated within a period of 12 months are considered as members of the group of vaccinated; this period is very long (1 year in the case of this pandemic), and the great length of this period should be more justified in the discussion, since the conditions with 12 months of evolution were very different.

The low number of people who responded to the survey is surprising, even with a prize money, what explanation do you have for the authors? It should be cited.

Throughout the work, there are several references to the greater appearance of side effects, in patients with chronic diseases, this association should be explained in more depth or include only those pathologies in which there is an unequivocal association between side effects and chronic pathologies.

The authors state that family doctors with more years of practice are more favorable to recommend vaccination, however, since they are vaccines with new technologies, it would seem reasonable that new doctors were the most prone to this vaccination recommendation. This fact should be commented on in the discussion.

Reviewer 3 Report

The manuscript describes an interesting approach in the general practitioner´s role in vaccination of the general population. It is clear that the medical personnel more acquainted with patients is more carrying and responsible regarding future complications. In general, the statistics are well done; however, there are some issues that would be interesting to discuss. 1) what is the knowledge the practitioners have on SARS CoV2 viral infection its complications and the unwanted effects of the vaccines? Which vaccine is more used in Hong Kong? Why most of the practitioners were vaccinated, but some of them did not recommend the vaccine? It is unclear if there are medical conditions related to the suggestion, Guillan Barre syndrome, cardiovascular disease autoimmunity.  Finally, it would be interesting to analyze the experience of the general practitioners in 1) incidence of SARS-CoV-2 infection in their patients, 2) the number of individuals reluctant to be vaccinated independent of medical counselling, 3) how many patients related to those practitioners died of the infection. 

There are minor issues, table 1, public, private practice and others? Please explain. Can the authors explain medical speciality? 
